# Prediction of Distant Metastases in Patients with Kidney Cancer Based on Gene Expression and Methylation Analysis

**DOI:** 10.3390/diagnostics13132289

**Published:** 2023-07-06

**Authors:** Natalya Apanovich, Alexey Matveev, Natalia Ivanova, Alexey Burdennyy, Pavel Apanovich, Irina Pronina, Elena Filippova, Tatiana Kazubskaya, Vitaly Loginov, Eleonora Braga, Andrei Alimov

**Affiliations:** 1Research Centre for Medical Genetics, 1 Moskvorechye St., Moscow 115522, Russia; apanovich2004@mail.ru (P.A.); eleonora10_45@mail.ru (E.B.); andrei.alimov2010@yandex.ru (A.A.); 2Federal State Budgetary Institution (N.N. Blokhin National Medical Research Center of Oncology) of the Ministry of Health of the Russian Federation, 24 Kashirskoe Shosse, Moscow 115478, Russia; a.matveevnmic@yandex.ru (A.M.); oncogen5@ronc.ru (T.K.); 3Institute of General Pathology and Pathophysiology, Baltijskaya St. 8, Moscow 125315, Russia; nata-i@list.ru (N.I.); burdennyy@gmail.com (A.B.); zolly_sten@mail.ru (I.P.); p.lenyxa@yandex.ru (E.F.); loginov7w@gmail.com (V.L.)

**Keywords:** clear cell renal cell carcinoma, metastasis, differential gene expression, CpG methylation

## Abstract

Clear cell renal cell carcinoma (ccRCC) is the most common and aggressive histological type of cancer in this location. Distant metastases are present in approximately 30% of patients at the time of first examination. Therefore, the ability to predict the occurrence of metastases in patients at early stages of the disease is an urgent task aimed at personalized treatment. Samples of tumor and paired histologically normal kidney tissue from patients with metastatic and non-metastatic ccRCC were studied. Gene expression was analyzed using real-time PCR. The level of gene methylation was evaluated using bisulfite conversion followed by quantitative methylation-specific PCR. Two groups of genes were analyzed in this study. The first group includes genes whose expression is significantly reduced during metastasis: *CA9*, *NDUFA4L2*, *EGLN3*, and *BHLHE41* (*p* < 0.001, ROC analysis). The second group includes microRNA genes: *MIR125B-1*, *MIR137*, *MIR375*, *MIR193A*, and *MIR34B/C*, whose increased methylation levels are associated with the development of distant metastases (*p* = 0.002 to <0.001, ROC analysis). Based on the data obtained, a combined panel of genes was formed to identify patients whose tumors have a high metastatic potential. The panel can estimate the probability of metastasis with an accuracy of up to 92%.

## 1. Introduction

The most common type of kidney cancer is a renal cell carcinoma (RCC) (approximately 90% of cases), of which 80% are clear cell renal cell carcinoma (ccRCC) [1,2]. Men are affected about twice as often as women, and a higher incidence occurs in the elderly population [3]. The disease progresses asymptomatically until late stages and is characterized by a high frequency of lethal outcomes, especially when metastasis develops. The mortality rate in the first year of observation is approximately 20% [4]. Approximately 30% of patients with localized disease develop distant metastases during follow-up [5]. The 5-year survival rate for ccRCC patients with metastases is 12% [1]. Significant progress in the treatment of metastatic ccRCC has been achieved in some patients due to the introduction of targeted immunotherapy, including the use of regimens aimed at suppressing the immune checkpoint [6]. Nevertheless, the problem of early detection of tumors with high metastatic potential remains relevant, and prognostic markers of metastasis need to be developed.

Currently, a large number of research groups are focused on studying the molecular mechanisms of metastasis and searching for molecular markers to predict its aggressiveness. The study of gene expression associated with kidney cancer metastasis is one such direction for identifying potential expression-based prognostic markers [7,8,9]. Recently, the level of post-transcriptional regulation, a significant part of which is carried out by microRNA (miRNA) molecules, has been actively investigated. In particular, such studies have included an analysis of gene methylation patterns involved in the carcinogenesis process [10,11]. Hypermethylation leading to inactivation of suppressor miRNA genes has been identified in various types of cancer, including ccRCC, and can be used as a biomarker to predict metastasis [12,13].

Despite the advances in this field, there are still not enough tests based on molecular markers in clinical practice to predict the risk of developing a metastasis [2]. The identification of such markers will create the opportunity to personalize approaches in treatment by forming groups for dynamic monitoring and timely detection of emerging metastases.

In this study, levels of expression of four protein-encoding genes and levels of methylation of nine miRNA genes were investigated in the same tumor samples, and their association with metastasis in ccRCC was analyzed. This made it possible to create a new panel of genes to assess the metastatic potential of kidney cancer tumors.

## 2. Materials and Methods

Samples of ccRCC tumor and normal tissue from the same organ obtained during surgical procedures were collected and clinically characterized at the N.N. Blokhin National Medical Research Centre of Oncology. After collection, the tissue was immediately frozen and stored at −70 °C. Only tumors from untreated patients were collected. The mean age of the patients was 60.5 ± 8.4 years at the time of diagnosis. Males prevailed among the patients—58.8%. All collected tumor samples were of the ccRCC subtype. The study used 80 paired samples of tumor and normal tissue. Distant metastases were found in 31 cases out of 80 at the time of surgical treatment. The criterion for inclusion in the study was the histological report of an examination of kidney tumor.

*Isolation of high molecular weight DNA.* DNA was isolated from tissue by phenol–chloroform extraction according to the standard protocol [14]. The quality and concentration of DNA was determined by optical density on a Nanodrop 1000 spectrophotometer (Thermo Fisher Scientific, Wilmington, DE, USA).

*Total RNA isolation.* RNeasy Mini Kit (QIAGEN, Germantown, MD, USA) was used for RNA isolation. Isolation was carried out according to the instructions for the kit. The quality of the isolated RNA was assessed by electrophoretic separation on a 1.8% agarose gel. The concentration of RNA in the aqueous solution was determined on a Nanodrop 1000 spectrophotometer (Thermo Fisher Scientific, Wilmington, DE, USA).

*Gene expression analysis.* The reverse transcription reaction was carried out using the ImProm-II™ Reverse Transcription System kit (Madison, WI, USA). Real-time polymerase chain reaction (PCR) was performed using Applied Biosystems (Foster City, CA, USA) USA kits: TaqMan^®^ Gene Expression Master Mix and TaqMan^®^ Gene Expression Assay, specifically designed for each analyzed gene (*GAPDH*, *CA9*, *NDUFA4L2*, *EGLN3*, and *BHLHE41*). Assay ID, respectively: Hs02758991_g1, Hs00154208_m1, Hs00220041_m1, Hs00222966_m1, and Hs00229146_m1. The *GAPDH* gene was used as an endogenous control. The relative mRNA expression level of the gene in tumor tissue compared to the expression level of the same gene in normal tissue of the same kidney was calculated using the ∆∆Ct (RQ) method with QuantStudio™ Design and Analysis Software v1.5.2.

*Analysis of miRNA gene methylation.* The level of miRNA gene methylation was analyzed using quantitative methylation-specific PCR with real-time detection (qMS-PCR) according to the method published in [15]. In brief, 1 µg DNA from the tumor and normal samples was used in bisulfite conversion. DNA conversion completeness was determined using the control locus *ACTB*, using oligonucleotides specific to the unconverted template. The commercial purified human genomic DNA #G1471 (Promega, Madison, WI, USA) was applied as a control for unmethylated alleles. The commercial enzymatically methylated human genomic DNA #SD1131 (Thermo Fisher Scientific, Waltham, MA, USA) was used as a positive control for 100% methylation. The set of miRNA gene specific primers has been described previously and applied in this study [16,17]. Primer sequences are as follows: m*MIR125B-1*F 5′-CGTTTTTTATTGAAATATTTCGTTTAG-3′, m*MIR125B-1*R 5′-CGAAACCCGCACAACCTCCT-3′, u*MIR125B-1*F *5′*-TTTTTGTTTTTTGTTTTTTATTGAAATA-3′, u*MIR125B-1*R 5′-CAAAACCCACACAACCTCCTATAAC-3′, m*MIR137*F 5′-GGTTTTTTGATTTTTTTCGGTGACG-3′, m*MIR137*R 5′-CCGCTAATACTCTCCTCGACTACGC-3′, u*MIR137*F 5′-GGTTTTTTGATTTTTTTTGGTGATGG-3′, u*MIR137*R 5′-CCCCCCTACCACTAATACTCTCCTCAA-3′, m*MIR375*F 5′-CGTCGTTATCGTTATCGTTATTTTAATC-3′, m*MIR375*R 5′-AATTTCTATTCTAAACCACGACCCC-3′, u*MIR375*F 5′-TTGTGTGTTGTTTTAGGGGAGATTTG-3′, u*MIR375*R 5′-ATAACTTACCCAAAACCATAAAAATCA-3′, m*MIR193A*F: 5′-GAGGTATTTGGTCGGAGCGTAC MR-3′, m*MIR193AR* 5′-GACCCCCGAAACCAACG-3′, u*MIR193A* F 5′-ATTGATTTATATTTTTGAGAGTGTTG-3′, u*MIR193A*R 5′-TCCCAAACTAACATACACTCCA-3′, m*MIR34B/C*F 5′-TTTAGTTACGCGTGTTGTGC-3′, m*MIR34B/C*R 5′-ACTACAACTCCCGAACGATC-3′, u*MIR34B/C*F 5′-TGGTTTAGTTATGTGTGTTGTGT-3′, u*MIR34B/C*R 5′-CAACTACAACTCCCAAACAATCC-3′, m*MIR1258*F 5′-AGGTCGTGGAAGTTATAGGC-3′, m*MIR1258*R 5′-CGAACCTACACCTAAACGC-3′, u*MIR1258*F 5′-ATTAGGTTGTGGAAGTTATAGGT-3′, u*MIR1258*R 5′-AACAAACCTACACCTAAACACA-3′, m*MIR107*F 5′-TGTGTAGTAGTTCGTTTATAGC-3′, m*MIR107*R 5′-GACTCTACGACTACTAAATCG-3′, u*MIR107*F 5′-TGTGTAGTAGTTTGTTTATAGTG-3′, u*MIR107*R 5′-CCAACTCTACAACTACTAAATC-3′, m*MIR132*F 5′-GCGTCGGCGTCGTTCG-3′ m*MIR132*R 5′-CGCCCCCGCCTCCTTCTA-3′, u*MIR132*F 5′-GTGTGTGTGTTGTTTG-3′ u*MIR132*R 5′-ACCCCCACCTCCTTCTAC-3′, m*MIR203A*F 5′-TTTCGGGTCGTGGAGGATTAGTC-3′, m*MIR203A*R 5′-ACTCCGAACGACGATAACCAACG-3′, u*MIR203A*F 5′-GTGGAGGATTAGTTGTGGGATTTAT-3′, u*MIR203A*R 5′-CCAACACAACAACACCTTTTATACAA-3′.

DNA amplification was performed using the qPCRmix-HS SYBR kit according to the manufacturer’s instructions on a Bio-Rad CFX96 Real-Time PCR Detection System (Bio-Rad, Hercules, CA, USA) according to the instructions provided with the device. The collected data were processed using the Precision Melt Analysis Software v 2.3 (Bio-Rad). The data were evaluated using a methylation index (MI) calculated for each sample based on the notion that MI is a continuous value ranging from 0 to 100% and can be interpreted as the percentage of methylation. MI = 0 indicates a complete absence of modification and MI = 100% indicates a complete methylation of the gene [18].

*Statistical data processing* was performed using Statistica 10.0 (StatSoft Inc., Tulsa, OK, USA). Differences were considered significant at *p* < 0.05. The Mann–Whitney U-test was used to search for correlations with metastasis. The Receiver Operator Characteristic (ROC) analysis was performed using the MedCalc software v 15.8 (MedCalc software Ltd., Ostend, Belgium). The cutoff values were used to determine the optimal marker system. An online calculator [19] was used to calculate the characteristics of the marker panel. Marker sets were evaluated by sensitivity, specificity, and the area under the curve (AUC). AUC values >0.7 were considered acceptable.

Since we conducted a study on the association between metastasis and the simultaneous expression of several genes, we applied the correction for the multiplicity of comparisons using the false discovery rate method (FDR) [20]. The application of this amendment avoids false “discoveries” that might arise for statistical reasons in multiple comparisons. The significance level was equal to or less than 0.05.

## 3. Results

A total of 80 paired tissue samples from patients with ccRCC were analyzed. Clinical and morphological characteristics of tumors in patients with ccRCC are presented in Table 1.

In all paired samples from patients with metastatic and non-metastatic ccRCC, levels of expression of protein-coding genes *CA9*, *NDUFA4L2*, *EGLN3*, and *BHLHE41* were determined. These genes were selected based on data from our previous study on the gene expression profiling of ccRCC [21]. Additionally, a group of miRNA genes (*MIR1258*, *MIR34B/C*, *MIR107*, *MIR132*, *MIR125B-1*, *MIR137*, *MIR375*, *MIR193A*, and *MIR203A*) was selected for analysis of the methylation levels and their association with ccRCC metastasis. The selection of these genes was based on previously published data [16]. The results of the analysis of *CA9*, *NDUFA4L2*, *EGLN3*, and *BHLHE41* expression and *MIR1258*, *MIR34B/C*, *MIR107*, *MIR132*, *MIR125B-1*, *MIR137*, *MIR375*, *MIR193A*, and *MIR203A* methylation levels are presented in Figure 1 and Table 2.

The study results showed that the reduction in expression levels of all four investigated genes, *CA9*, *NDUFA4L2*, *EGLN3*, and *BHLHE41*, is associated with tumor metastasis (Table 2). The relationship between methylation levels of miRNA genes and metastasis was selective. wherein an increase in methylation levels of all significantly different genes indicates the metastatic potential of the tumor. The best values were obtained for five out of nine genes, *MIR125B-1*, *MIR137*, *MIR375*, *MIR193A*, and *MIR34B/C*.

According to the results obtained, an analysis of the possibility of using the identified genes as potential markers of metastasis was made.

For further study, the following genes were selected: *CA9*, *NDUFA4L2*, *EGLN3*, *BHLHE41*, *MIR125B-1*, *MIR137*, *MIR375*, *MIR193A*, *MIR34B/C*, and *MIR1258*. Calculated U-test values and the results of logistic regression analysis were applied as selection criteria.

ROC analysis was used to analyze the particularities of the relationship between gene expression and methylation levels with the metastatic potential of the tumor (Table 3, Figure 2).

The *CA9*, *NDUFA4L2*, *EGLN3*, and *BHLHE41* genes were found to be the most relevant to the panel for predicting metastasis. 

Among miRNA genes, *MIR125B-1*, *MIR137*, *MIR375*, *MIR193A*, and *MIR34B/C* have the best classifier qualities (Table 2 and Table 3, Figure 2). 

An unfavorable prognosis for the development of metastases is a decrease in the expression level of *CA9*, *NDUFA4L2*, *EGLN3*, and *BHLHE41* and an increase in the level of methylation of miRNA genes *MIR125B-1*, *MIR137*, *MIR375*, *MIR193A*, and *MIR34B/C.*


The values of prognostic sensitivity and specificity for these genes ranged from 65% to 97% and from 49% to 88%, respectively (Table 3).

Consequently, the differences in median values, logistic regression method, and ROC analysis indicate an association between the nine genes studied and RCC metastasis. 

Based on AUC values >0.7, nine genes were identified as the strongest candidates for biomarkers of metastasis.

To improve the sensitivity and specificity indices, and to increase the practical significance of the obtained data, three sets of markers were compiled, with each analyzed as a separate group of genes (Table 4). The prognostic significance of these marker sets was evaluated for future use in molecular genetic testing of tissue samples of clear cell renal cell carcinoma (Table 4).

Figure 3 shows the graphical result of the ROC analysis.

Simultaneous analysis of mRNA expression levels of *CA9*, *NDUF4L2*, *EGLN3*, and *BHLHE41* genes and/or methylation levels of miRNA genes (*MIR125B-1*, *MIR137*, *MIR375*, *MIR193A*, and *MIR34B/C*) increases the reliability of the method. The decrease in the probability of random association of the marker with ccRCC metastasis indicates an improvement in the reliability of the method in this case.

In the practical use of a panel consisting only of protein-coding genes, a recorded decrease in the expression of three out of four of them indicates a high probability of developing metastases. When testing a panel consisting of only five miRNA genes, the recorded increased methylation of four of them indicates a high probability of metastasis. Both judgments are based on the limit value determined by ROC analysis for each gene. When testing a combined panel of genes, any combination of the six or more recorded events listed above corresponds to a high metastatic potential of the tumor.

The corresponding calculation based on the obtained data shows that the combined panel based on simultaneous measurement of gene expression levels and methylation levels has the best sensitivity (87%) and specificity (95%), making it the most reliable predictive panel. However, in case of limited resources, performing either only gene expression level measurements or only methylation level measurements also shows fairly good sensitivity and specificity.

The presented data suggest that the developed combined panel has high sensitivity and specificity. Our determined AUC level of 0.915 indicates high accuracy in predicting metastasis development, which is another important argument in favor of using the combined panel in molecular genetic testing. It should also be noted that the probability of absence of metastasis in case of a negative test is 92%.

## 4. Discussion

The prediction of metastasis in patients with localized and locally advanced clear cell kidney cancer can be of great practical importance, since it can affect both the volume of surgical intervention, the need for lymphadenectomy, and the need for adjuvant immunotherapy. In cases where renal tumors do not have metastases, the standard scope of surgical intervention is nephrectomy without regional lymph node dissection. Conducting a tumor biopsy prior to surgery or using non-invasive methods of obtaining biological material in the future, such as liquid biopsy, may make it possible to identify a group of patients with a high risk of developing metastases based on the results of gene expression analysis and methylation. In these patients, lymphadenectomy can improve long-term treatment outcomes and provide better staging. Furthermore, prediction of metastasis may be the key to deciding whether adjuvant therapy is necessary. In particular, the randomized phase 3 study KEYNOTE-564 [22] showed a benefit of pembrolizumab (PDL-1 inhibitor) in the adjuvant regimen in patients at intermediate/high risk of progression in terms of recurrence-free survival. It was shown that pembrolizumab reduced the risk of metastasis by 32% with the adjuvant regimen. The two-year disease-free survival rate was 77.3% in the pembrolizumab group and 68.1% in the placebo group (hazard ratio for recurrence or death 0.68; 95% CI 0.53–0.87, *p* = 0.002). More accurate prediction of metastases based not only on clinical signs, but also on the expression and methylation levels of the genes studied, will help to more selectively identify patients with a high risk of progression and, as a result, to more effectively prescribe adjuvant therapy. To meet this challenge, our study identified a set of genes that predict the development of metastases in ccRCC through the assessment of mRNA expression levels of *CA9*, *NDUF4L2*, *EGLN3*, and *BHLHE41*, and/or methylation levels of miRNA genes (*MIR125B-1*, *MIR137*, *MIR375*, *MIR193A*, and *MIR34B/C*) in the tumor.

All four protein-encoding genes (*CA9*, *NDUF4L2*, *EGLN3*, and *BHLHE41*) are direct targets of HIF1 (hypoxia-inducible factor 1) [23,24,25,26]. Under conditions of normal oxygenation and in the absence of mutation in the *VHL* gene, the product of this gene is a part of the E3 ubiquitin ligase complex, which promotes the attachment of ubiquitin to hydroxylated transcription factors HIFs, leading to their degradation by the proteasome pathway. Under hypoxic conditions, HIF1α translocates to the nucleus where it dimerizes with the constitutively expressed HIF1b, forming the HIF1 complex. Accumulation of HIF1α also occurs due to inactivation of the *VHL* gene. Under hypoxia and/or absence of functional pVHL, the VHL complex does not bind to the non-hydroxylated transcription factors HIFs, leading to their accumulation in cells and the formation of the HIF1 complex [27,28]. HIF1 activates the transcription of a number of target genes by binding to the hypoxia response element (HRE) located in the regulatory region of the target gene. Induction of the expression of genes activated by hypoxia is necessary for cellular adaptation to a microenvironment with low oxygen levels, for example, during the growth of solid tumors and the transition of homeostatic regulation to a new level (Figure 4A) [29].

The increase in expression levels of the genes we have identified as associated with metastasis in ccRCC is apparently mediated by HIF1 processes of adaptation to a microenvironment with low oxygen levels, as well as by the accumulation of HIF1α caused by the inactivation of the *VHL* gene. However, during tumor development, their expression levels decrease. It can be assumed that as the kidney tumor progresses, processes related to a strong initial response to hypoxia, regulated by the action of HIF1α, diminish. Meanwhile, other processes such as inflammation develop.

CA9 (carbonic anhydrase 9) is a transmembrane glycoprotein. Its function is to regulate the pH of the cell and maintain the acid–base balance in the body. CA9 catalyzes the reversible hydration of carbon dioxide to bicarbonate, allowing tumor cells to maintain a neutral pH level inside the cell, while acidifying the extracellular microenvironment [30]. CA9 expression has been studied in some detail in ccRCC, and it is molecularly linked to pVHL and regulated by the transcription factor HIF1α [31,32]. Decreased CA9 immunohistochemical staining intensity in metastases compared to corresponding primary tumor samples was demonstrated by Bui et al. [33]. It was also shown that decreased CA9 expression occurs in tumors with the highest malignant potential [33]. A meta-analysis conducted by Zhao Z. et al. [34] showed by qPCR that low *CA9* expression levels correlate with a number of clinical characteristics, such as a high degree of differentiation, presence of distant metastases and metastases to lymph nodes. A correlation with decreased expression level and low disease-specific, progression-free, and overall survival was also demonstrated [34]. Some researchers associate decreased CA9 expression in patients with poor prognosis with the activation of AKT and mTOR pathways, which makes further tumor growth less dependent on hypoxia and shifts it to an alternative pathway [32].

The decrease in CA9 expression levels, according to our results, indicates an increased metastatic potential of ccRCC consistent with the published data.

The product of the *NDUFA4L2* gene is a subunit of NADH dehydrogenase, which is a component of the mitochondrial respiratory chain. NDUFA4L2 has a pronounced dependence on hypoxia and is a direct target of HIF1α. The functions of NDUFA4L2 are poorly understood, but likely involve the regulation of oxidative phosphorylation through interaction with subunits within complex I of the mitochondrial respiratory chain. Activated under hypoxic conditions, NDUFA4L2 reduces mitochondrial oxygen consumption by inhibiting complex I activity, limiting intracellular ATP production under low oxygen conditions [35]. Decreased expression of the *NDUFA4L2* gene is associated with a disruption of the molecular mechanism of cell transition to anaerobic glycolysis, which prevents the reduction of reactive oxygen species (ROS) production and promotes further progression [36]. The significance of *NDUFA4L2* expression for the development of ccRCC was first shown by us [37], and confirmed later in other studies [21,38,39,40]. In the study by Meng et al., NDUFA4L2 was found to be overexpressed in non-small cell lung cancer (NSCLC) tissue and cell lines under hypoxic conditions. Activated by HIF1a, NDUFA4L2 suppressed mitochondrial ROS production in NSCLC cells. Knockdown of NDUFA4L2 promoted increased ROS production, apoptosis, and promotion of the epithelial–mesenchymal transition (EMT) of NSCLC cell lines [41]. It is likely that a decrease in the expression level of *NDUFA4L2* leading to the activation of EMT is also the mechanism of its effect on metastasis in ccRCC.

EGLN3 (PHD3) is a prolyl hydroxylase 3 whose expression is HIF1-dependent and contributes to preventing apoptosis in cells under hypoxic stress conditions. There is also feedback in this signaling pathway. Under normal conditions, PHD3 hydroxylates HIF1a, which then binds to VHL, becomes ubiquitinated, and degraded via the proteasome. PHD3 also hydroxylates and activates the HIF1a coactivator PKM2 [42]. Another important function of PHD3 is its involvement in regulating glucose metabolism. It has been shown that PHD3 regulates key glycolytic enzymes, including PFKP, TPI1, ENO1, PGAM1, and LDHA, together with the glucose transporter GLUT1. LDHA catalyzes the conversion of pyruvate to lactate. Thus, PHD3 maintains a high rate of glycolysis and lactate production in cancer cells [43]. LDHA is overexpressed in many types of cancer and plays a crucial role in tumor proliferation, invasion, and metastasis [44]. A decreased level of *EGLN3* expression leads to a disruption of the hydroxylation of extracellular signal-regulated kinase 3 (Erk3)—one of the key players in regulating tumor progression [45]. The expression of this gene has also been found to be associated with overall survival [45] and progression [46] in kidney cancer. Screening studies have shown differential expression of EGLN3 in cancer cells [47,48]. 

BHLHE41 (DEC2) is a basic helix–loop–helix (bHLH) transcription factor that is activated by hypoxia [26]. Under hypoxic conditions, HIF-1α induces the transcription of BHLHE41. BHLHE41 can in turn block the hypoxic response by promoting proteasomal degradation of HIF-1α [44]. Among the many functions of BHLHE41 are participation in cell differentiation, immune response, regulation of molecular clocks, and carcinogenesis [49]. Significant upregulation of *BHLHE41* expression in kidney cancer was observed in TCGA analysis [50]. The importance of increased expression of this gene for disease progression is currently not well understood, but it has been shown that its suppression in cultured cells by siRNA leads to inhibition of proliferation [51]. In another study, transfection of BHLHE41 into cultured cancer cells resulted in an increase in the proportion of cells in S and G2 phases compared to those in the G1 phase [52]. Therefore, increased expression of BHLHE41 can enhance the proliferative properties of tumor cells at the early stage of ccRCC. At the same time, increased expression of BHLHE41 can inhibit tumor development by suppressing invasion, as it is demonstrated in various types of cancer [49,53,54,55]. We found that the known mechanisms of action of the BHLHE41 correspond to the relationship between the expression level of this gene and ccRCC metastasis.

Thus, from a theoretical point of view, our findings suggest that the association between the reduced expression of studied genes and metastasis allows us to judge the degree of activation of molecular mechanisms of tumor progression in the cell.

DNA methylation is necessary for normal development and transcriptional regulation, but in cancer, it is usually altered towards hypermethylation. In our study, we identified five miRNA genes (*MIR125B-1*, *MIR137*, *MIR375*, *MIR193A*, and *MIR34B/C*) whose hypermethylation is associated with metastasis in ccRCC. Increased frequency of methylation of these genes may indicate their tumor suppressor function or the activation of expression of target protein-encoding genes (oncogenes).

The role of miR-125b in cancer progression is contradictory, as it can act both as a tumor suppressor and as an oncogene in different tumor types. MiR-125b originates from miR-125b-1 and miR-125b-2 and leads to degradation of target mRNA or inhibition of translation by binding to the 3’-untranslated regions (3’-UTR) of target mRNA. MiR-125b can promote signal transduction in various signaling pathways [56]. MiR-125b may exert its tumor suppressor function by directly interacting with the delta-catalytic subunit of phosphoinositide 3-kinase (PIK3CD) and reducing its expression, as shown in the anaplastic thyroid cancer cell line (ATC). Overexpression of PIK3CD leads to increased migration and invasion of ATC cells. In addition, exogenous miR-125b reduced the expression of PI3K, p-Akt, and p-mTOR in ATC cells [57]. In hepatocellular carcinoma, miR-125b has an inhibitory effect on EMT and EMT-related features through SMAD2 and SMAD4, which are signal transducers of the TGFβ signaling pathway [58]. Increased expression of miR-125b inhibited the invasion and migration of ovarian cancer cells and was associated with decreased expression of EIF4EBP1 (eukaryotic translation initiation factor 4E-binding protein 1). EIF4EBP1 is an oncogene that plays a critical role in controlling protein synthesis, cell growth, and survival, thereby promoting oncogenesis [59]. Direct correlation between decreased expression levels of miR-125b and increased gene *MIR125B-1* methylation levels have been shown in various types of cancer [60,61,62]. Our results indicate that hypermethylation of *MIR125B-1* in ccRCC samples with metastasis suggests tumor suppressor properties of this miRNA in kidney cancer.

Overexpression of miR-137 in RCC cells significantly suppressed cell proliferation, migration, and invasion, and induced apoptosis in vitro, as well as inhibited tumor growth in vivo. Additionally, miR-137 was shown to inhibit the activation of the PI3K/AKT signaling pathway in renal cancer cell lines [63]. The direct target of miR-137 in RCC cells is PIK3R3, which is involved in regulating the AKT/mTOR signaling pathway. By suppressing the expression of PIK3R3, miR-137 inhibited cell migration and invasion in RCC [64]. Similar results were shown in gastric cancer, where miR-137 suppressed the activation of the PI3K/AKT signaling pathway by targeting cyclooxygenase-2 (Cox-2), both in vitro and in vivo [65]. Hypermethylation of the *MIR137* promoter was shown in gastric cancer tissue, and its suppression induced activation of its target, Cdc42, which is associated with cancer initiation and progression [66]. MiR-137 was shown to directly target Snail and inhibit EMT in ovarian cancer, which is an early and critical stage of metastasis [67]. MiR-137 expression was suppressed and correlated with hypermethylation in renal cancer [68]. Thus, our data on the relationship between hypermethylation of *MIR137* and tumor metastasis are consistent with the existing published data.

Overexpression of miR-375 suppressed migration and invasion and inhibited proliferation by inducing apoptosis in renal cancer cell lines. Its suppressive effect was achieved by inhibiting PDK1 and preventing AKT phosphorylation in renal cancer cells [69]. Similar results were shown in pancreatic cancer cells, where miR-375 suppressed cell growth and induced apoptosis by negatively regulating the expression of 3-phosphoinositide-dependent protein kinase 1 (PDK1) [70]. Activation of miR-375 inhibited migration and invasion of human non-small cell lung carcinoma (NSCLC) cells by directly targeting the human epidermal growth factor receptor 2 (HER-2) [71]. Suppression of HER-2 expression by miR-375 has also been found in gastric cancer [72]. It has also been demonstrated that miR-375 suppresses growth, metastasis, and drug sensitivity of ovarian cancer cells [73]. The expression of miR-375 was decreased in samples with altered methylation in ccRCC [68].

Increased expression of miR-193a-3p/5p inhibited migration, invasion, and EMT of (NSCLC) cells in vitro, as well as lung metastasis formation in vivo. Furthermore, ERBB4 and S6K2 were identified as direct targets of miR-193a-3p, while PIK3R3 and mTOR were direct targets of miR-193a-5p in NRML cells. It was also shown that miR-193a-3p/5p can inactivate the AKT/mTOR signaling pathway [74]. The expression of miR-193a inhibited growth, oncogenicity, and radiation sensitivity of medulloblastoma cells with increased expression of MYC. MAX, DCAF7, and STMN1 were identified as novel targets of miR-193a, which may contribute to its anti-tumor effect. The expression of miR-193a in medulloblastoma cells leads to widespread repression of gene expression, including genes involved in WNT signaling, NOTCH signaling, cell cycle regulators, and DNA replication, as well as chromatin organization and modification [75]. It has been shown that the product of the *MIR193A* gene is one of the key post-transcriptional regulators of protein-7 expression, an adaptor protein of the growth factor receptor, which is one of the key mediators involved in receptor tyrosine kinase signaling. Aberrant elevation of GRB7 levels is often associated with the progression of human cancer. Reduction of miR-193a-3p expression due to DNA hypermethylation is a dynamic process of ovarian cancer progression [76]. Serum levels of miR-193a-3p were significantly elevated in stage I pancreatic cancer patients compared to cancer-free controls (*p* < 0.01) [77].

Members of the miR-34 family have been described as tumor suppressors in various types of cancer. In cervical cancer, the expression of miR-34b was decreased, and its expression level was associated with increased malignant potential. Overexpression of miR-34b strongly suppressed cell proliferation and induced apoptosis in cervical cancer cell lines [78]. The expression of miR-34b in colorectal adenocarcinoma tissue was negatively correlated with the expression of p-PI3K, p-AKT, and mTOR proteins. MiR-34b can inhibit colorectal adenocarcinoma [79]. It has also been shown that miR-34b/c suppresses CDK4/6 expression in breast cancer cell lines [80]. CpG hypermethylation of *MIR34B* suppresses miR-34b in prostate cancer. The anti-proliferative and anti-migratory/invasive effects of miR-34b were partially due to the suppression of the AKT pathway and EMT markers. It has also been shown that miR-34b inhibits oncogenicity both in vitro and in vivo [81]. MiR-34b/c may be suppressed in gastric cancer by hypermethylation of *MIR34B/C*. On the other hand, miR-34b/c regulates the expression of genes involved in the p53-mediated signaling network, suppressing cell proliferation, migration, and metastasis [82]. MiR-34b/c enhanced the attachment of cancer cells and suppressed cell growth and invasion in a mouse model of lung cancer. Moreover, patients with lung adenocarcinoma had better survival rates with higher levels of miR-34a/b/c than those with lower levels [83].

According to our results, hypermethylation of the miRNA genes *MIR125B-1*, *MIR137*, *MIR375*, *MIR193A*, and *MIR34B/C* is associated with the metastatic potential of tumors. From the published data, it has been shown that methylation of these genes is associated with decreased expression of the corresponding miRNAs. Taken together, these data may indicate a suppressive function of miR-125b-1, miR-137, miR-375, miR-193a, and miR-34b/c in RCC. From the data described in publications, the studied miRNAs are involved in many signaling pathways and some of them exert their regulatory functions, both as suppressors and as oncogenes. The common feature of these five miRNAs is their involvement in the regulation of the PI3K/AKT/mTOR signaling pathway. As is known, the PI3K/AKT/mTOR signaling pathway is often activated in kidney cancer and leads to increased cell growth, proliferation, and metastasis [84]. This can explain the unidirectional effect of hypermethylation of the identified miRNA genes on the progression of RCC (Figure 4B).

A distinctive feature of our proposed approach for predicting metastasis in kidney cancer patients is the combination of analyzing the expression and methylation of two different groups of genes, protein-coding and miRNA, in one study. Currently, there are no prognostic gene panels constructed based on this principle, highlighting the originality of our proposed approach. By using this methodology, gene groups that function independently in two different pathways are involved, which is registered as a realization of decreased expression or increased methylation in individual samples, complementing each other as markers, thereby contributing to increased prognostic accuracy during testing.

Most of the nomograms used in the clinic are based on the use of indirect signs related to disease progression. The main factors for the prognosis of metastasis that are currently used are TNM classification, G grade of tumor differentiation, C reactive protein, neutrophil-to-lymphocyte ratio, and several other clinical signs in different prognostic models [2]. To determine the independent value of our gene panel for improving existing prognostic models, a multivariate analysis is planned, taking into account the above clinical data on a large cohort of patients with a long follow-up period. We believe that our proposed approach, based on direct registration of genetic events associated with metastases, can improve the accuracy and quality of prognosis because it is based on direct assessment.

## 5. Conclusions

Determining significant markers of metastasis in ccRCC is a priority area of research, as there are currently no confirmed markers. All studies on this issue are exploratory and represent separate results that have not yet been integrated into a system. The data presented above are conceptual in nature and expand existing ideas about methodological approaches for creating new prognostic gene panels for assessing the metastatic potential of a kidney tumor.

The proposed gene panel with high probability allows the prediction of the development of metastases based on the analysis of the expression levels of *CA9*, *NDUFA4L2*, *EGLN3*, and *BHLHE41* genes, and the methylation of miRNA genes *MIR125B-1*, *MIR137*, *MIR375*, *MIR193A*, and *MIR34B/C*. We believe that the suggested set of markers is useful in planning an extended analysis of a small amount of tumor tissue in order to improve molecular testing for predicting the development of renal metastases.

## Figures and Tables

**Figure 1 diagnostics-13-02289-f001:**
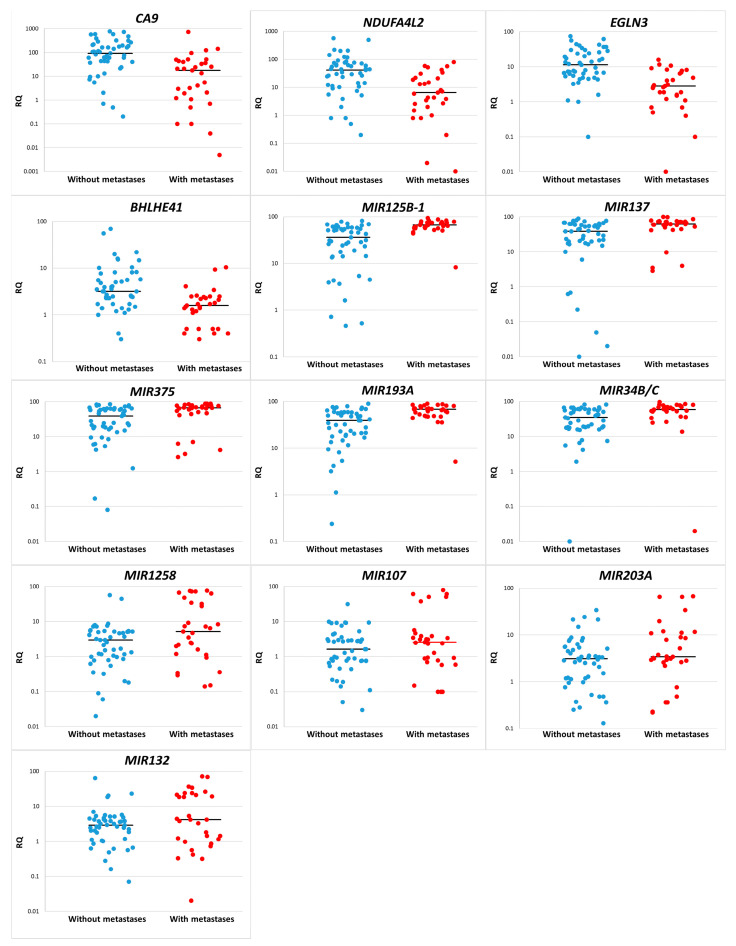
Relative gene expression and methylation levels (RQ) in groups without metastasis (●) and with metastasis (●). Gene expression values and methylation levels are presented on a logarithmic scale. The median is indicated by a line on the graph.

**Figure 2 diagnostics-13-02289-f002:**
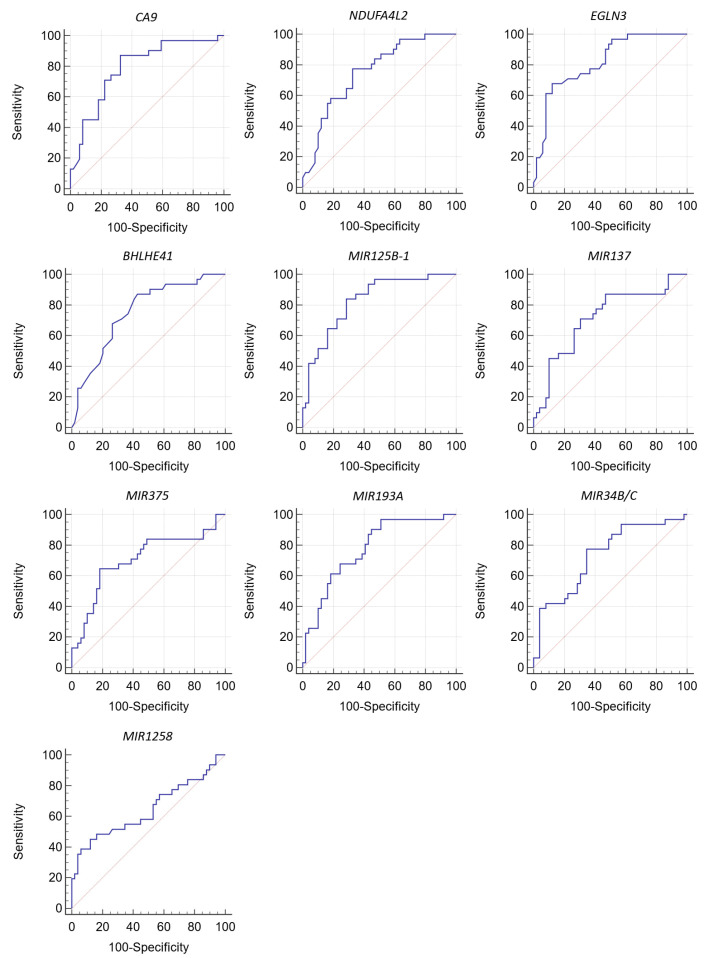
ROC curve analysis to test the association of gene expression and methylation levels with ccRCC metastasis.

**Figure 3 diagnostics-13-02289-f003:**
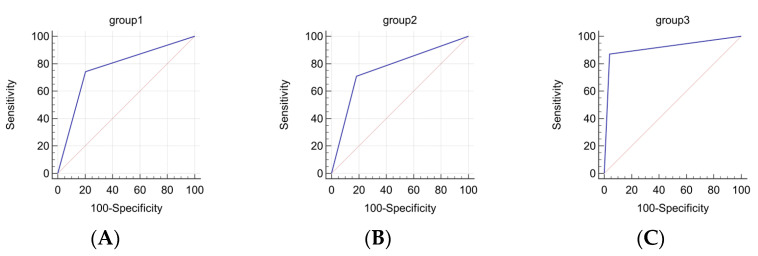
ROC analysis of marker panels: (**A**) *CA9*, *NDUFA4L2*, *EGLN3*, and *BHLHE41*; (**B**) *MIR125B*-1, *MIR137*, *MIR375*, *MIR193A*, and *MIR34B*/C; (**C**) *CA9*, *NDUFA4L2*, *EGLN3*, *BHLHE41*, *MIR125B*-1, *MIR137*, *MIR375*, *MIR193A*, and *MIR34B*/C.

**Figure 4 diagnostics-13-02289-f004:**
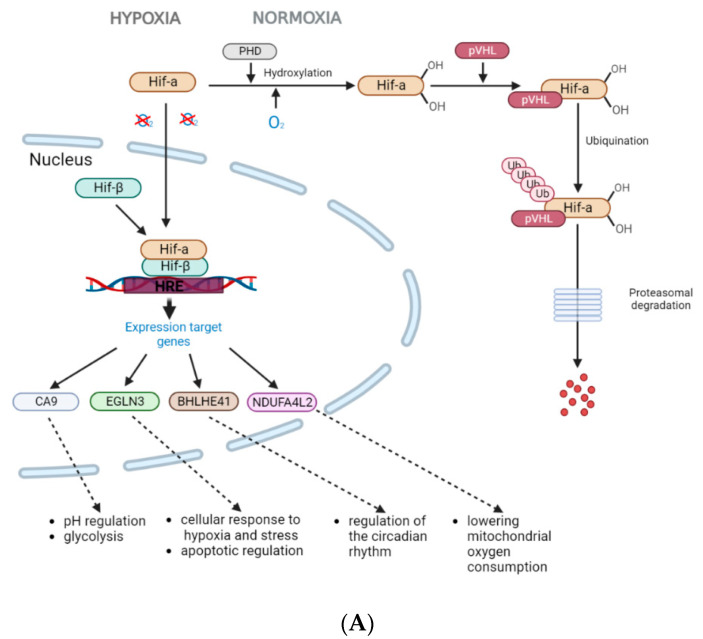
Schematic representation of gene interaction pathways and their regulation in cancer. (**A**). HIF1 regulation of gene expression under hypoxia. (**B**). miRNAs regulating the PI3K/AKT signaling pathway in cancer.

**Table 1 diagnostics-13-02289-t001:** Clinical and morphological characteristics of tumors in patients with ccRCC.

Characteristic		Number of Samples	Age	Gender M/F (%M/%F)
TNM stage	I	20	60.7 ± 11.1	10/10 (50/50)
	II	6	59.3 ± 8.2	5/1 (83.3/16.7)
	III	23	62.3 ± 6.9	16/7 (69.6/30.4)
	IV	31	59.3 ± 7.7	16/15 (51.6/48.4)
The presence of metastases	With distant metastases	31	59.3 ± 7.7	16/15 (51.6/48.4)
	No metastases	49	61.3 ± 8.9	31/18 (63.3/36.7)
Localization of distant metastases	Lungs	19	60.4 ± 8.3	9/10 (47.4/52.6)
	Adrenal	9	56.2 ± 8.1	5/4 (55.6/44.4)
	Bones	3	61.0 ± 6.6	1/2 (33.3/66.7)
	Other	4	57.3 ± 7.0	4/0 (100/0)

**Table 2 diagnostics-13-02289-t002:** Median values of gene expression (A) and methylation (B) levels, and significance of differences in ccRCC tumor groups.

Gene	The Median Value		(Mann–Whitney U-Test), *p* =	Logistic Regression, *p* =
	In the Non-Metastasis Group	In the Metastasis Group		
(A)				
*CA9*	92.7	17.8	<0.001	0.022
*NDUFA4L2*	41.1	6.5	<0.001	0.007
*EGLN3*	11.4	2.8	<0.001	0.004
*BHLHE41*	3.2	1.6	<0.001	0.018
(B)				
*MIR125B-1*	36.27	66.34	<0.001	0.001
*MIR137*	38.10	61.84	0.002	0.006
*MIR375*	38.99	66.19	0.003	0.007
*MIR193A*	38.99	67.58	<0.001	0.001
*MIR34B/C*	35.26	59.23	0.001	0.004
*MIR1258*	2.97	5.19	0.040	0.010
*MIR107*	1.62	2.57	0.252	0.036
*MIR203A*	3.11	3.42	0.235	0.061
*MIR132*	2.9	4.17	0.252	0.036

Note: significance levels (*p*) are presented taking into account the correction of the Benjamini–Hochberg method.

**Table 3 diagnostics-13-02289-t003:** The association gene expression and methylation levels with metastasis in ccRCC.

Gene	Area under ROC Curve (AUC)	95% CI	Cutoff Value	Significance Level, *p* (Area = 0.5)	Sensitivity	Specificity
*CA9*	0.789	0.684–0.873	≤51.3 *	<0.001	87.10	67.35
*NDUFA4L2*	0.753	0.644–0.842	≤22 *	<0.001	77.42	67.35
*EGLN3*	0.818	0.716–0.895	≤4.2 *	<0.001	67.74	87.76
*BHLHE41*	0.751	0.642–0.841	≤2.6 *	<0.001	87.10	57.14
*MIR125B-1*	0.827	0.726–0.902	>55.18 **	<0.001	83.87	71.43
*MIR137*	0.716	0.604–0.811	>57.62 **	0.001	70.97	69.39
*MIR375*	0.706	0.593–0.802	>64.29 **	0.001	64.52	81.63
*MIR193A*	0.776	0.668–0.861	>34.65 **	<0.001	96.77	48.98
*MIR34B/C*	0.732	0.621–0.825	>50.35 **	<0.001	77.42	65.31
*MIR1258*	0.644	0.529–0.748	>7.15 **	0.033	45.16	87.76

Note: *—expression level; **—methylation level; significance levels (*p*) are presented taking into account the correction of the Benjamini–Hochberg method.

**Table 4 diagnostics-13-02289-t004:** Characterization of combined prognostic panels based on protein-coding genes and miRNA genes.

Gene Group	Sensitivity/ Specificity	Area under ROC Curve (AUC)	Significance Level, *p* (Area = 0.5)	Negative Predictive Value % (95% CI)	Positive Predictive Value % (95% CI)
*CA9* *NDUFA4L2* *EGLN3* *BHLHE41*	74.19/79.59	0.769	<0.0001	82.98 (69.19–92.35)	69.70 (51.29–84.41)
*MIR125B-1* *MIR137* *MIR375* *MIR193A* *MIR34B/C*	70.97/81.63	0.763	<0.0001	81.63 (67.98–91.24)	70.97 (51.96–85.78)
*CA9* *NDUFA4L2* *EGLN3* *BHLHE41* *MIR125B-1* *MIR137* *MIR375* *MIR193A* *MIR34B/C*	87.10/95.92	0.915	<0.0001	92.16 (81.12–97.82)	93.10 (77.23–99.15)

## Data Availability

The data presented in this study are available on request from the corresponding author.

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
