# Peer review of "Prediction of Distant Metastases in Patients with Kidney Cancer Based on Gene Expression and Methylation Analysis"

_diagnostics, 2023, doi:10.3390/diagnostics13132289_

Round 1

Reviewer 1 Report

1. The paper presents a valuable contribution to the prediction of distant metastases in kidney cancer patients using gene expression and methylation analysis. However, a more thorough discussion on the clinical implications of the findings would enhance the relevance of the study.   2. The methodology employed in this study appears robust, but further details on the specific gene expression and methylation analysis techniques used would be beneficial for better understanding and reproducibility.   3. The paper should provide information on the characteristics of the patient cohort used for analysis, including the number of patients, their demographics, and clinical outcomes. This information is crucial for assessing the generalizability of the findings.   4. The statistical analyses conducted in the study are comprehensive. However, it would be helpful to include additional details on the statistical significance thresholds and adjustments applied to account for multiple testing.   5.  It would be valuable to include independent validation on an external dataset to assess the generalizability of the findings.   6. The paper lacks information on the specific computational tools or software packages used for data analysis and model development.   7. The clinical implications of the proposed prediction model should be clearly discussed, including its potential impact on patient management and personalized treatment strategies.   8. The discussion section should address the limitations of the study, such as potential sources of bias, and suggest future directions to overcome these limitations and further improve the prediction model.   9. The clinical utility of the proposed prediction model could be enhanced by providing insights into its sensitivity, specificity, positive predictive value, and negative predictive value, particularly in comparison to existing approaches.   10. The paper should discuss the potential impact of confounding factors, such as tumor stage and treatment history, on the identified gene expression and methylation markers. This would help assess their independent predictive value.

Moderate editing of English language required

Author Response

Thank you very much for useful recommendations to improve the article.
All of these are accounted for:
1. Clinical implications added to the discussion section
2. Missing information has been added to the Materials and Methods section to better understand the results, including more details on specific methods for gene expression and methylation analysis.
3. Information about patient cohort characteristics has been added.
4. The false discovery rate (FDR) has been added and described in methods.
5. We agree that such validation can be conducted. However, there is still no consensus among clinicians on the management and treatment regimens of patients with kidney cancer. Therefore, it is not possible to obtain all the necessary data for such testing in the necessary volume.
6. Information on specific computational tools has been added to the "Materials and Methods" section
7. A discussion of the potential use of the predictive model and examples of its possible application in patient management has been added to the Discussion section.
8. Limitations in the use of the results and ways to improve them are discussed in the Discussion section.
9. Missing information about the sensitivity and specificity of the panel discussed has been added to the text of the article, see Table 4
10. Missing information about the clinical status of the patients has been added to the text of the article, Materials and Methods section. At the time of sample collection, all patients were not receiving any treatment.

Reviewer 2 Report

Dear Authors, 

I would like to congratulate you for the submission of a well prepared manuscript. While my remarks are going to be minor, I do consider them to be important in benefiting the reader understand your paper.

The abstract is quite well structured.

The introduction is comprehensive while concise.

- However, it would be maybe useful to remove the 1st sentence. This is a well known statement and it makes for an abrupt dive into the subject.

- "The study of genes whose expression is associated"  could be rephrased as: "The study of gene expression associated"

- line 49 "microRNA" -typo

- line 54 "to predict the risk of developing a metastasis" - or another rephrase

- Last two paragraphs are the same. Please choose one. Personally, i prefer the latter. 

Materials and Methods: 

- Please move Table 1 to results section!

- line 82 "spectrophotometrically" - could you please provide the Brand and model of the machine? - I would assume a nanodrop method was used

- Please correct the section "Analysis of microRNA gene methylation" so that names are not in between "" and so that it is written in the same style as the other sections which are quite well structured 

- line 112 - please remove the link and reference it as bibliography

Results section is okay, but should include Table 1 and it would be advisable that the patient data for each group is summarised. Please at least include average of age and gender distribution for each stage in TNM and metastasis. 

The Discussion and Conclusion sections are okay!

I am looking forward to seeing the final version of the manuscript. 

Best of luck and best regards, 

Your Reviewer

Author Response

Thank you very much for useful recommendations to improve the article.

All of these are accounted for:

- The 1st sentence has been removed.

- The sentence has been rephrased.

- line 49 "microRNA" – corrected.

- line 54 the sentence has been rephrased.

- The duplicate paragraph has been removed.

- Table 1 has been moved to the results section.

- The brand and model of the spectrophotometer were listed.

- The section "Analysis of microRNA gene methylation" has been revised and corrected.

- line112 The link has been moved to the bibliography section.

- Missing data on the distribution of patients by age and sex for each stage of TNM, as well as data on distant metastases, are added to the text of the article, Table 1.

Round 2

Reviewer 1 Report

The authors have addressed my comments. The paper is now much improved.  Therefore, my suggestion is to accept it.  

 Minor editing of the English language is required